# UNCONDITIONAL IMAGE-TEXT PAIR GENERATION WITH MULTIMODAL CROSS QUANTIZER

**Hyungyung Lee, Sungjin Park, Edward Choi**
KAIST, Republic of Korea
{ttumyche, zxznm, edwardchoi}@kaist.ac.kr

## ABSTRACT

Though deep generative models have gained a lot of attention, most of the existing works are designed for the unimodal generation task. In this paper, we explore a new method for unconditional image-text pair generation. We propose MXQ-VAE, a vector quantization method for multimodal image-text representation. MXQ-VAE accepts a paired image and text as input, and learns a joint quantized representation space, so that the image-text pair can be converted to a sequence of unified indices. Then we can use autoregressive generative models to model the joint image-text representation, and even perform unconditional image-text pair generation. Extensive experimental results demonstrate that our approach effectively generates semantically consistent image-text pair and also enhances meaningful alignment between image and text.

## 1 INTRODUCTION

Deep generative models are an active research area, but the main research focus has been unimodal generation, either unconditional (GAN (Goodfellow et al., 2014), VAE (Kingma & Welling, 2013), PixelCNN (Van Oord et al., 2016), GPT (Brown et al., 2020)) or conditional (Oscar (Li et al., 2020), DALL-E (Ramesh et al., 2021)). As of yet, unconditional multimodal generation has not been much explored. One previous work (Shimanuki, 2019) proposes generating image and text simultaneously with a GAN-based approach. However, the core idea was to treat text as an image, where the model generates a pair of images, one for the picture and one for the text, thus this process must undergo the OCR process (Smith, 2007).

To overcome this limitation, we propose Multimodal Cross Quantizer VAE (MXQ-VAE) for unconditional image-text pair generation and enhancing the semantic consistency of the generated image-text pair. More specifically, we adopt a vector quantization method (Oord et al., 2017) and Transformer encoder (Vaswani et al., 2017) that learns a joint quantized representation space. Thus, we can convert the image-text pair to a sequence of unified indices, and train Autoregressive Transformer (Brown et al., 2020) over this unified indices sequence, allowing for generating semantically consistent image-text pair simultaneously without any conditional input.

In order to show the effectiveness of our approach, we design several baselines and construct the corrupted data, denoted as the Degree dataset, by gradually adjusting the degree of the alignment between image and text. The experimental results demonstrate that our approach achieves consistent results on the Degree dataset, while baselines fail, and also can generate semantically consistent image-text pair over baselines. The contributions of our work can be summarized as follows:

- For the first time, we propose MXQ-VAE, a vector quantization method that learns a joint quantized representation space for unconditional image-text pair generation
- The quantitative and qualitative results show the effectiveness of our approach over baselines with considerable semantic consistency of the generated image-text pair on text-augmented MNIST, denoted as Caption MNIST and Oxford Flower-102 (Nilsback & Zisserman, 2008).
- Additionally, The experimental result on the Degree dataset demonstrates that our approach learns meaningful alignment between image and text compared to non-MXQ module model.

## 2 RELATED WORKS

**Image and Text Generation**   Most image and text generation studies focus on unimodal generation. (Li et al., 2020; Ramesh et al., 2021) propose image-to-text and text-to-image generation respectively. (Kingma & Welling, 2013; Goodfellow et al., 2014; Van Oord et al., 2016) propose unconditional image generation task and (Brown et al., 2020) generates text. (Shimanuki, 2019), on the other hand, proposes unconditional image-text generation, but it generates text of an image and must undergo the OCR (Smith, 2007).

**Vector Quantized Variational Autoencoder**   VQ-VAE (Oord et al., 2017) is first proposed to represent images into the discrete representation. (Razavi et al., 2019) enhances the VQ-VAE by using multi-scale hierarchical encoder. (Ramesh et al., 2021) utilizes gumbel-softmax relaxation to better optimize the VQ-VAE.

**The Transformer Family**   The Transformer (Vaswani et al., 2017) is an encoder-decoder architecture. (Chen et al., 2020b; Zhou et al., 2020; Li et al., 2020) utilize Transformer encoder to jointly learn imagetext representation. (Brown et al., 2020) is an autoregressive language model based on Transformer decoder. Recently, by factorizing image and video into the discrete representation, (Brown et al., 2020) also can model the continuous data and has shown remarkable results (Chen et al., 2020a; Yan et al., 2021).

## 3 METHOD

Figure 1 illustrates our approach. Following DALL-E, we adopt a two-stage approach.

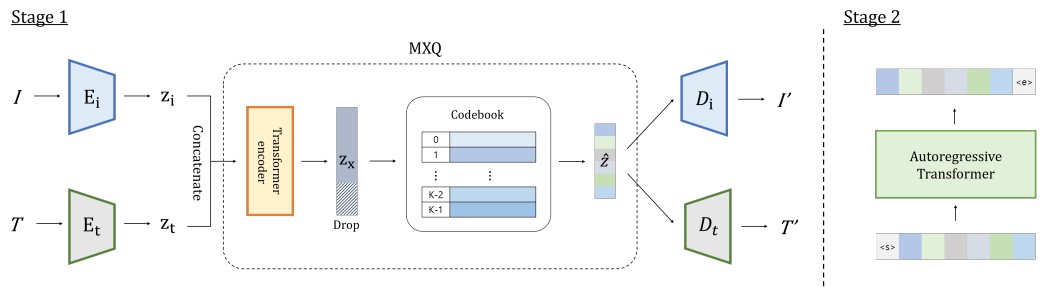

Figure 1: **Framework of unconditional image-text pair generation.** Left: MXQ-VAE takes a paired image and text as input, and learns a joint quantized representation space, then converts the image-text pair to a sequence of unified indices. Right: Autoregressive Transformer models the joint image-text representation with the unified indices sequence. At inference, MXQ-VAE decodes the sampled sequences to image-text pair

### 3.1 STAGE 1: LEARN A JOINT QUANTIZED REPRESENTATION SPACE

Inspired by VQ-VAE, we design MXQ-VAE with CNN-based encoders, decoders (2D CNN for image and 1D CNN for text), linear layers, Transformer encoder and codebook $C \in \mathbb{R}^{K \times D}$, where $K$ and $D$ are the size and dimension of $C$.

- Given an input image $I \in \mathbb{R}^{H \times W \times 3}$ and text $T \in \mathbb{R}^{w \times d}$, each encoder $E_i$ and $E_t$ encodes the input into a set of feature maps $z_i \in \mathbb{R}^{H' \times W' \times D}$ and $z_t \in \mathbb{R}^{w' \times D}$, where $H, W$ and $w$ are height, width of image and length of text, respectively, and are downsampled to $H', W'$ and $w'$ by a factor of 8, $d$ is a dimension of the word embedding.

- To learn a joint quantized representation space, we design a MXQ module with Transformer encoder and codebook $C$. Transformer encoder takes the concatenation of $z_i$ and $z_t$, then outputs joint representation. For more deeper representation, we drop the the part of the sequence $z_t$ from the joint representation then the remaining part $z_x \in \mathbb{R}^{L \times D}$, where $L = H' \times W'$, is discretized onto a joint quantized representation space by searching a nearest neighbor in codebook $C$ and produces a sequence of unified indices $\hat{z} \in \mathbb{R}^{L \times D}$.

- Each $D_i$ and $D_t$ in the Figure 1 consists of a linear layer and a decoder (i.e. reversed encoder). To ensure the decoder takes desired size as input, the linear layer is applied to the spatial dimension (i.e. $L$) of the $\hat{z}$ then the decoder reconstructs the original input from the reduced $\hat{z}$.

## 3.2 STAGE 2: UNCONDITIONAL IMAGE-TEXT PAIR GENERATION WITH A UNIFIED SEQUENCE

We adopt Autoregressive Transformer to model the joint representation with the sequence of unified indices. During training, MXQ-VAE encodes image and text to the unified sequence and then Autoregressive Transformer is trained to predict the next index of the unified sequence. At inference, MXQ-VAE decodes the sequence sampled from the model to image-text pair.

## 4 EXPERIMENTS

### 4.1 DATASET

**1) Caption MNIST** Based on (Shin et al., 2021), we build 600k image-text pairs, each image-text pair contains several color, digit and position (Figure 2 in Appendix). we have 4 colors (white, red, green and blue), 10 digits (0 to 9) and 5 positions (center, top left, top right, bottom left and bottom right). According to the filled quadrant, we called each pair as Single and Quad1 to 4. Single image-text pairs only have a color and digit, this colored digit is located in the center of the image .
**2) Degree Dataset** We construct the Degree dataset based on Caption MNIST by gradually adjusting the degree of the alignment between image and text (Figure 2 in Appendix). More specifically, we replace the color and digit in text to other random color and digit. For instance, the Quad4 can have 5 degrees range from totally unpaired (Degree 0) to originally paired (Degree 4).
**3) Oxford Flower-102** contains 81,890 images of 102 flower species, with 10 captions per image.

### 4.2 BASELINES

All baselines also generate image-text pair without any conditional input, and there is only a difference in which modality comes first.
**1) Only Sharing $e$.** At Stage 1, this baseline only shares the codebook $e$ without any module to jointly combine both image and text. At Stage 2, depending on which modality comes first, we refer to it as I&T_VQ or T&I_VQ.
**2) Separately Quantizing.** At Stage 1, two distinct models quantize each modality separately. For image, it follows original VQ-VAE. There are two versions for text, (1) it also follows VQ-VAE; (2) since text is originally discrete data, we directly use the word embedding. At Stage 2, depending on which modality comes first, in case (1), we refer to it as I_T_VQ or T_I_VQ, and in case (2), we refer to it as I(VQ)_T(Embd) or T(Embd)_I(VQ).
**3) MXQ-VAE (w/o text compression).** We change the $E_t$ and $D_t$ of MXQ-VAE to the Transformer encoder. In this model, the text encoder outputs the same number of embeddings as the input.

### 4.3 RESULTS

For the details of evaluation metrics, please refer to Appendix A.1.

We first study **the effect of the MXQ module with respect to the alignment between image and text**. We evaluate all models on the Degree dataset. Table 1 and Table 4, 5, 6, and 7 in Appendix show the results. We observe that Only Sharing $e$ and MXQ-VAE (w/o text compression) can not fully capture the alignment between image and text. In fact, MXQ-VAE (w/o text compression) totally fails to learn the relationship between image and text. Our approach, on the other hand, shows the best approximation, close to 1.0, 0.75, 0.5, 0.25 and 0.0 on Quad4. Also, we examine the advantage of the sequence drop (*i.e.* dropping $z_t$) that brings in considerable improvement.

We measure **the semantic consistency of the generated image-text pairs** on Caption MNIST. Table 2 shows the performance of all models. We can observe that MXQ-VAE outperforms all baselines on every quadrant. Also note that all baselines are vulnerable to which modality is given first to generate the image-text pair. All models given text first significantly degrade up to 27% (in I_T_VQ and T_I_VQ) on average compared to the model given image first. We assume that this is

due to the fact that quantized image is longer than text and image often contains more complex information than text. Our MXQ-VAE, on the other hand, alleviates this issue with a unified sequence. The generated image-text pairs (Figure 3 in Appendix) show the high fidelity image and paired text.

Table 1: Alignment accuracy on Quad4 Degree dataset.

| Models | Quad4 | | | | |
|---|---|---|---|---|---|
| | Degree 4 | Degree 3 | Degree 2 | Degree 1 | Degree 0 |
| MXQ-VAE | 0.978 | 0.734 | 0.489 | 0.247 | 0.002 |
| Only Sharing $e$ | 0.486 | 0.443 | 0.394 | 0.358 | 0.315 |
| MXQ-VAE (w/o dropping $z_t$) | 0.902 | 0.669 | 0.436 | 0.244 | 0.066 |
| MXQ-VAE (w/o text compression) | 0.997 | 0.993 | 0.99 | 0.984 | 0.98 |

Table 2: Semantic consistency of the generated image-text pairs on Caption MNIST.

| Models | Single | Quad1 | Quad2 | Quad3 | Quad4 | Average |
|---|---|---|---|---|---|---|
| MXQ-VAE | **0.992** | **0.992** | **0.99** | **0.995** | **0.939** | **0.982** |
| I&T_VQ | 0.979 | 0.926 | 0.675 | 0.434 | 0.255 | 0.654 |
| T&I_VQ | 0.803 | 0.78 | 0.458 | 0.282 | 0.161 | 0.497 |
| I_T_VQ | 0.975 | 0.935 | 0.93 | 0.872 | 0.627 | 0.868 |
| T_I_VQ | 0.816 | 0.794 | 0.647 | 0.477 | 0.253 | 0.598 |
| I(VQ)_T(Embd) | 0.953 | 0.953 | 0.956 | 0.958 | 0.849 | 0.945 |
| T(Embd)_I(VQ) | 0.086 | 0.895 | 0.913 | 0.916 | 0.828 | 0.728 |
| MXQ-VAE (w/o text compression) | 0.976 | 0.974 | 0.979 | 0.982 | 0.873 | 0.957 |

We also assess our approach on Oxford Flower-102. Table 3 shows the brief result, refer to Table 8 in Appendix for the whole result. Compared with our baseline model Joint GAN (Shimanuki, 2019), MXQ-VAE outperforms in all scores, indicating its capability to generate semantically consistent flower image-text pairs. This result shows that our approach can extend to more general domain in the future. Figure 4 and 5 in Appendix show the generated image-text pairs by MXQ-VAE and Joint GAN respectively. As we can see, MXQ-VAE generates more realistic image and high quality text. In summary, quantitative and qualitative results demonstrate that our approach enhances the meaningful alignment between image and text, and it also encourages to generate semantically consistent image-text pairs.

Table 3: Sentence similarity of the generated image-text pairs on Oxford Flower-102. * means we measure the score with reported image-text pairs in Joint GAN (Shimanuki, 2019).

| Models | Sentence Similarity ($\uparrow$) | | |
|---|---|---|---|
| | Top-1 | Top-5 | Top-10 |
| MXQ-VAE | **0.967** | **0.958** | **0.952** |
| Joint GAN (Shimanuki, 2019) * | 0.832 | 0.813 | 0.802 |

## 5 CONCLUSION

In this study, we first propose MXQ-VAE, a vector quantization method for unconditional image-text pair generation that quantize the paired image and text to a joint quantized representation space. Extensive experimental results demonstrate the effectiveness of our approach. In future work, we plan on extending our approach to a large-scale complex dataset.

## ACKNOWLEDGEMENTS

This work was supported by Institute of Information & Communications Technology Planning & Evaluation (IITP) grant (No.2019-0-00075, Artificial Intelligence Graduate School Program(KAIST)) and National Research Foundation of Korea (NRF) grant (NRF-2020H1D3A2A03100945) funded by the Korea government (MSIT), and the KAIST-NAVER Hyper-Creative AI Center.

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

# A APPENDIX

## A.1 EVALUATION METRICS

- **Semantic parser** Following (Shin et al., 2021), we evaluate the semantic consistency between the generated image and text on Caption MNIST. We extract a set of position, color, and digit with the rule-based semantic parser of the generated text. With a color and digit classifier trained with the image of Caption MNIST and achieved 100% and 99.5% accuracy each, we predict the color and digit of the part of the generated image corresponding to the position that exist in the generated text. Then, we measure whether both predicted color and digit math or not at that position.

- **Alignment accuracy** It measures the alignment of image and text using Degree dataset. For instance, in Quad1, the Degree 1 input text is 'the green 0 is on the upper right.' and the Degree 0 input text is 'the white 1 is on the upper right.'. Input image always depicts the original text, Degree 1 in this case. If the reconstructed text is 'the green 0 is on the upper right.' for both Degree 1 and 0 input text, the alignment accuracy will be 1 and 0 for each degree. With the well-alignment space, the model should figure out the corrupted part and does not reconstruct that part as it is. According to this, in Quad3, the desired accuracy would be 1, 0.66, 0.33 and 0 for each degree.

- **Sentence similarity** To evaluate the semantic consistency of the generated image-text pairs on Oxford Flower-102 (Nilsback & Zisserman, 2008), we measure a sentence similarity between the generated text and the pooled original text that has the same class. Specifically, we first train a shallow CNN on the original image to classify the 82 categories and achieves 99% accuracy. We predict the class of the generated image with this classifier and pool all the original text that has the same class. The pre-trained BERT (Devlin et al., 2018) takes the generated text paired with that image and the pooled original text respectively and outputs each vector by mean pooling operation. We calculate the cosine similarity between the vector of the generated text and the others and report a Top-1, 5 and 10 averaged score. To validate this metric, we conduct on original Oxford Flower-102 and achieve 0.999, 0.963 and 0.962 average similarity score respectively.

## A.2 IMPLEMENTATIONS

- **Stage 1** We use 2D-, 1D-CNN encoder and decoder for images and text respectively. For the Caption MNIST, the embedding latent space $e$ is $256 \times 128$, the input size is $64 \times 64 \times 3$ image and $64 \times 128$ word embedding, and then each input is downsampled to $8 \times 8 \times 128$ and $8 \times 128$. For Oxford Flower-102, the embedding latent space $e$ is $512 \times 64$, the input size is $224 \times 224 \times 3$ image and $80 \times 128$ word embedding. Each input is downsampled to $28 \times 28 \times 64$ and $20 \times 64$. We split text into word based on white space. For MXQ-VAE (w/o text compression) model, we use Transformer encoder with 2 layers, 2 attention heads and 128 embedding dimension. We train model with 1 NVIDIA RTX A6000 with 800 and 496 batch size for Caption MNIST and Oxford Flower-102 respectively. Following (Oord et al., 2017), Our model optimizes using the following objectives:

$$L = ||I - D_i(\hat{z}_i)||_2^2 + ||T - D_t(\hat{z}_t)||_2^2 + ||sg[z_x'] - e||_2^2 + \beta||sg[e] - z_x'||_2^2 \qquad (1)$$

  The loss term consists of reconstruction losses for image and text, a codebook loss, and a commitment loss. $sg$ refers to a stop-gradient. The commitment loss is weighted by a hyperparameter $\beta$ and we set $\beta$ to 0.25 and 0.5 for Caption MNIST and Oxford Flower-102 respectively.

- **Stage 2** We adopt (Brown et al., 2020) for an autoregressive generative model with 8 layers, 8 attention heads and 512 embedding dimension. The model optimizes negative log-likelihood loss. We train model with 1 NVIDIA RTX A6000 with 800 and 40 batch size for Caption MNIST and Oxford Flower-102.

In all our experiments, we use AdamW (Loshchilov & Hutter, 2017) with $\beta_1 = 0.9$, $\beta_2 = 0.99$, and the cosine learning rate scheduler (Loshchilov & Hutter, 2016) from $5e - 4$ to $0$ multiplied by the batch size.

| Quadrant | Image | Text |
|---|---|---|
| Single | | this is white 7. |
| Quad1 | | the 6 on the lower right is blue. |
| Quad2 | | the green 0 is on the upper right, and the top left 8 is red. |
| Quad3 | | the lower left 4 is blue, the red 0 is on the lower right, and the 2 on the top right is green. |
| Quad4 | | the bottom left 1 is blue, the upper right 7 is white, the bottom right 0 is red, and the upper left 3 is green. |

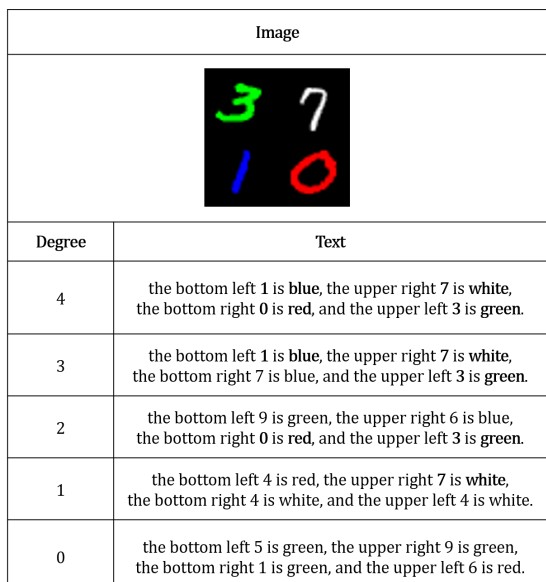

| Degree | Text |
|---|---|
| 4 | the bottom left **1** is **blue**, the upper right **7** is **white**, the bottom right **0** is **red**, and the upper left **3** is **green**. |
| 3 | the bottom left **1** is **blue**, the upper right **7** is **white**, the bottom right 7 is blue, and the upper left **3** is **green**. |
| 2 | the bottom left 9 is green, the upper right 6 is blue, the bottom right **0** is **red**, and the upper left **3** is **green**. |
| 1 | the bottom left 4 is red, the upper right **7** is **white**, the bottom right 4 is white, and the upper left 4 is white. |
| 0 | the bottom left 5 is green, the upper right 9 is green, the bottom right 1 is green, and the upper left 6 is red. |

Figure 2: An example of Caption MNIST (Left) and Degree Dataset (Right).

Table 4: Alignment accuracy on Single Degree dataset. The desired accuracy would be 1 and 0.

| Models | Single | |
|---|---|---|
| | Degree 1 | Degree 0 |
| MXQ-VAE | 0.99 | 0.0 |
| Only Sharing $e$ | 1.0 | 0.84 |
| MXQ-VAE (w/o dropping $z_t$) | 0.995 | 0.064 |
| MXQ-VAE (w/o text compression) | 1.0 | 0.972 |

Table 5: Alignment accuracy on Quad1 Degree dataset. The desired accuracy would be 1 and 0.

| Models | Quad1 | |
|---|---|---|
| | Degree 1 | Degree 0 |
| MXQ-VAE | 0.994 | 0.0 |
| Only Sharing $e$ | 1.0 | 0.772 |
| MXQ-VAE (w/o dropping $z_t$) | 0.995 | 0.0 |
| MXQ-VAE (w/o text compression) | 1.0 | 0.981 |

Table 6: Alignment accuracy on Quad2 Degree dataset. The desired accuracy would be 1, 0.5 and 0.

| Models | Quad2 | | |
|---|---|---|---|
| | Degree 2 | Degree 1 | Degree 0 |
| MXQ-VAE | 0.995 | 0.498 | 0.0 |
| Only Sharing $e$ | 0.905 | 0.776 | 0.65 |
| MXQ-VAE (w/o dropping $z_t$) | 0.993 | 0.41 | 0.04 |
| MXQ-VAE (w/o text compression) | 1.0 | 0.993 | 0.988 |

Table 7: Alignment accuracy on Quad3 Degree dataset. The desired accuracy would be 1, 0.66, 0.33, and 0.

| Models | Quad3 | | | |
|---|---|---|---|---|
| | Degree 3 | Degree 2 | Degree 1 | Degree 0 |
| MXQ-VAE | 0.993 | 0.662 | 0.331 | 0.001 |
| Only Sharing $e$ | 0.713 | 0.64 | 0.558 | 0.483 |
| MXQ-VAE (w/o dropping $z_t$) | 0.991 | 0.634 | 0.309 | 0.066 |
| MXQ-VAE (w/o text compression) | 1.0 | 0.995 | 0.99 | 0.988 |

Table 8: Sentence similarity of the generated image-text pairs on Oxford Flower-102. * means we measure the score with reported image-text pairs in Joint GAN (Shimanuki, 2019).

| Models | Sentence Similarity ($\uparrow$) | | |
|---|---|---|---|
| | Top-1 | Top-5 | Top-10 |
| MXQ-VAE | **0.967** | **0.958** | **0.952** |
| I&T_VQ | 0.942 | 0.931 | 0.925 |
| T&I_VQ | 0.925 | 0.912 | 0.903 |
| I_T_VQ | 0.92 | 0.91 | 0.904 |
| T_I_VQ | 0.941 | 0.929 | 0.922 |
| I(VQ)_T(Embd) | 0.96 | 0.949 | 0.942 |
| T(Embd)_I(VQ) | 0.934 | 0.919 | 0.911 |
| MXQ-VAE (w/o text compression) | 0.964 | 0.955 | 0.949 |
| Joint GAN (Shimanuki, 2019) * | 0.832 | 0.813 | 0.802 |

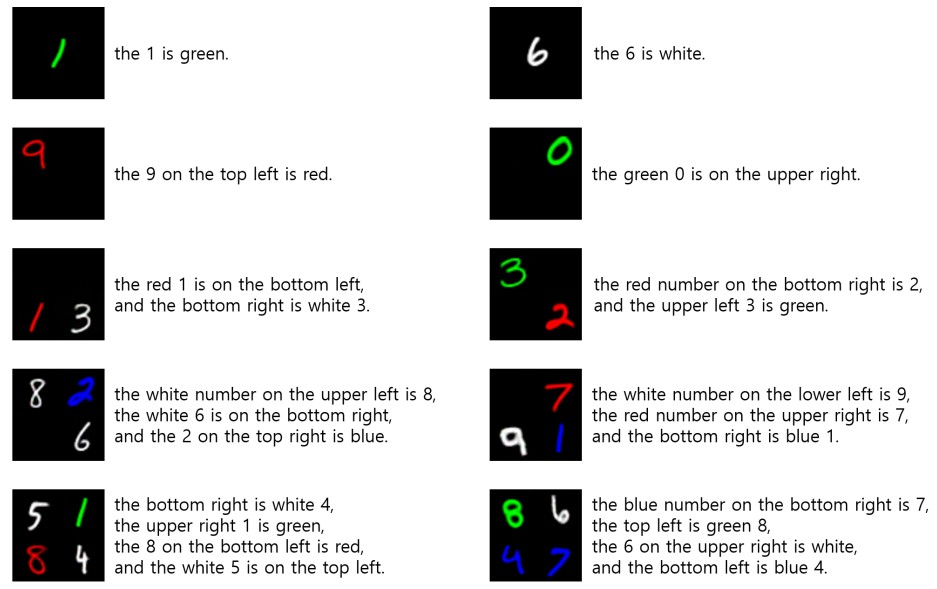

Figure 3: Generated image-text pairs of MXQ-VAE on Caption MNIST

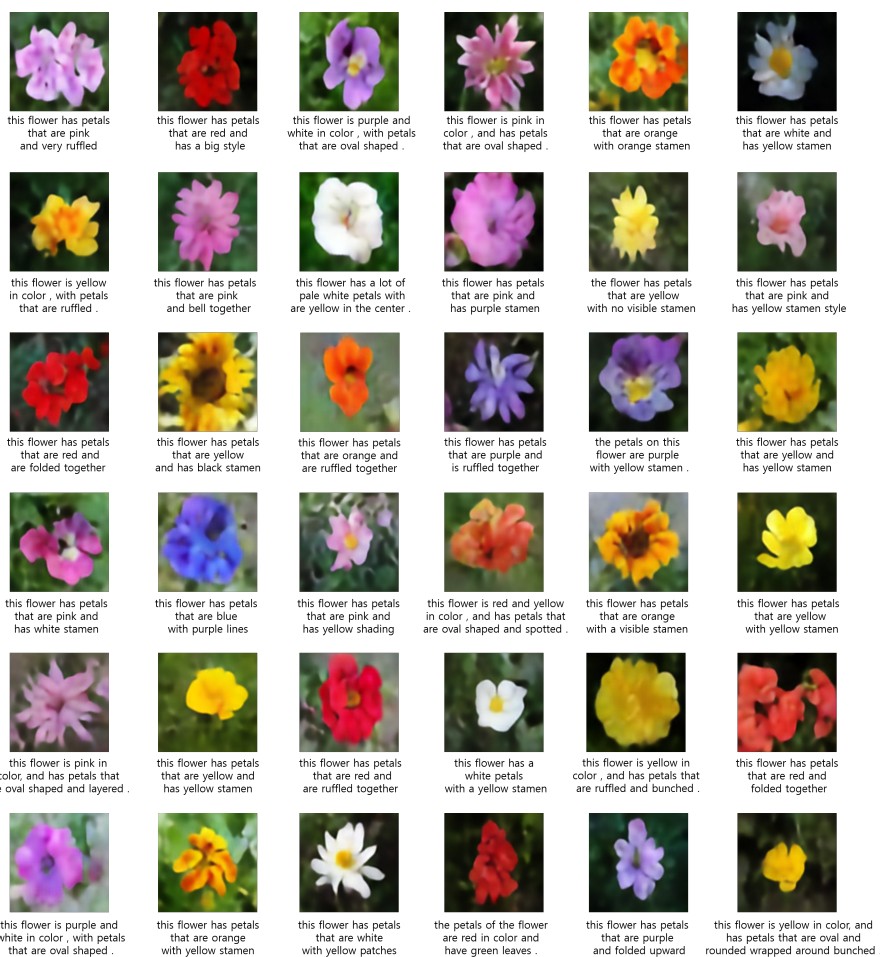

Figure 4: Generated image-text pairs of MXQ-VAE on Oxford Flower-102

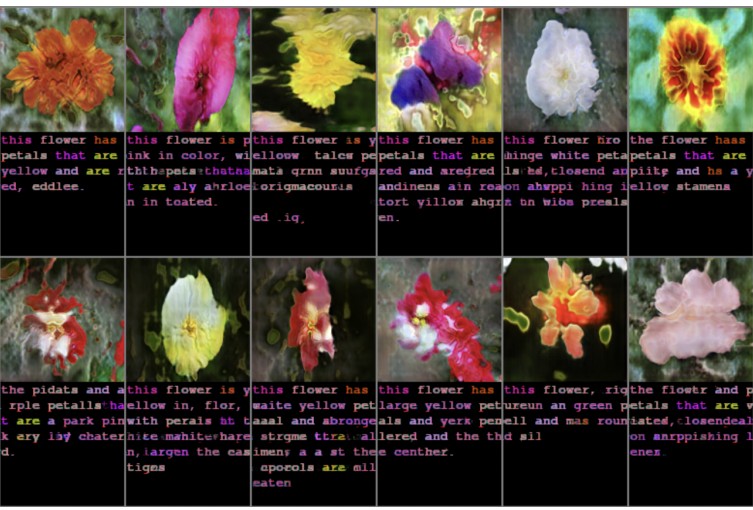

Figure 5: Generated image-text pairs from Joint GAN (Shimanuki, 2019) on Oxford Flower-102

