# OpenReview forum: "UNCONDITIONAL IMAGE-TEXT PAIR GENERATION WITH MULTIMODAL CROSS QUANTIZER"
_ICLR.cc/2022/Workshop/DGM4HSD — ICLR 2022 DGM4HSD workshop Poster_

### Official Review · Reviewer_nkMy · 2022-03-13
**Interesting Approach to Multimodal Generation**

**Rating:** 7
**Confidence:** 4

**Review:**

&nbsp;

## **SUMMARY**

&nbsp;

The authors present a method MXQ-VAE for joint image-text representation and use this for unconditional joint image-text generation. The empirical evaluation appears to be sound and has some interesting findings that will no doubt stimulate discussion at the workshop. As such, I recommend acceptance with the following minor points the authors may wish to consider.

&nbsp;

## **MINOR POINTS**

&nbsp;

1. On page 1, "To overcome this limitation, we propose Multimodal Cross Quantizer VAE (MXQ-VAE) for unconditional image-text pair generation, not text of an image, and enhancing the semantic consistency of the generated image-text pair.", it's slightly unclear to me what is meant by "not text of an image"?

2. On page 1, "In order to show the effectiveness of our approach, we design several baselines and construct the corrupted data, denoted as Degree dataset, by gradually adjusting the degree of the alignment between image and text." It would be good to explain exactly why this shows the effectiveness of the proposed approach, in other words what is the exact motivation for constructing the Degree dataset?

3. In the introduction (or in the appendix if space is an issue!) it would be nice to include some example application domains of joint image and text generation e.g. in the art world it may be used to generate artwork together with a textual description of its meaning [1] (or recover lost artwork [2]) using a combination of images and textual descriptions.

4. Typo in the abstract, "image text pairs" in place of "image text pair".

5. What is a "sequence of unified indices"?

6. In the Related work section under the Vector Quantized Variational Autoencoder, there is a typo in "to better optimize" in place of "to better optimizing".

7. I appreciate the authors may be tight on space but it may be beneficial to define the meaning of the notation H, W and w in section 3.1.

8. It may be beneficial to add a related work section in the appendix on joint image-text representations e.g. [3].

9. Why is the alignment accuracy of MXQ-VAE (w/o text compression) close to 1 in tables 1 and 2? This seems to contradict what is said about alignment accuracy in appendix A.1?

10. If the authors made their code available the paper would be further strengthened.

&nbsp;

## **REFERENCES**

&nbsp;

[1] Kell et al. Extracting associations and meanings of objects depicted in artworks through bi-modal deep networks. Electronic Imaging 2022.

[2] Bourached et al. Recovery of underdrawings and ghost-paintings via style transfer by deep convolutional neural networks: A digital tool for art scholars. Electronic Imaging, 2021(14), pp.42-1.

[3] Chen et al. Uniter: Learning universal image-text representations. ECCV 2020.

&nbsp;

---

### Official Review · Reviewer_KM67 · 2022-03-28
**Interesting paper with good results**

**Rating:** 7
**Confidence:** 4

**Review:**

**Summary:**

This paper proposes a new method for unconditional image-text pair generation based on a vector quantization method in the latent space.  The authors show with detailed experiments that the proposed method can achieve good results.

**Strength**
1. The paper is well written and easy to follow.
2. The experiments are detailed with good results.

**Limitation**
1. Transformer encoder and vector quantization approaches have been proposed and studied by previous works. Concatenating the feature of text and image, modifying the model to take the concatenation as the input and generate image-text output might not be a very novel contribution given existing works on latent variable models for text and images.
2. The proposed framework has a very specific setting: it requires large-scale paired image-text training data, which can be expensive to collect in real-world applications. The proposed framework might not be able to be applied to more commonly seen applications where paired image-text training data are not available.

---

### Decision · Program_Chairs · 2022-03-26

Accept (Poster)